# A Systematic Review on the Biomechanics of Breakfall Technique (Ukemi) in Relation to Injury in Judo within the Adult Judoka Population

**DOI:** 10.3390/ijerph19074259

**Published:** 2022-04-02

**Authors:** Ruqayya Lockhart, Wiesław Błach, Manuela Angioi, Tadeusz Ambroży, Łukasz Rydzik, Nikos Malliaropoulos

**Affiliations:** 1Centre for Sports and Exercise Medicine, William Harvey Research Institute, Queen Mary University of London, London E1 4DG, UK; r.lockhart@smd17.qmul.ac.uk (R.L.); m.angioi@qmul.ac.uk (M.A.); contact@sportsmed.gr (N.M.); 2Faculty of Sport, University School of Physical Education in Wroclaw, 51-612 Wroclaw, Poland; wieslaw.judo@wp.pl; 3Institute of Sports Sciences, University of Physical Education, 31-571 Krakow, Poland; tadek@ambrozy.pl; 4Sports Clinic, Rheumatology Department, Barts Health NHS Trust, London E1 4DG, UK; 5Sports and Exercise Medicine Clinic, 4639 Thessaloniki, Greece

**Keywords:** sports injuries, judo, Ukemi

## Abstract

Objectives: To investigate the biomechanics of Ukemi in relation to head and neck injury in adult judokas with varying skill sets. Design: Narrative systematic review. Methods: An extensive literature search was performed using PubMed, Google Scholar, Science direct and EMBASE from inception to April 2021. Studies were included if they: (1) reported biomechanical analysis of judo throws and Ukemi; (2) were on adult judoka populations; (3) discussed injury related to judo technique. The included studies were assessed for risk of bias using a five-part modified STROBE checklist. A narrative synthesis was performed due to the heterogeneity of included studies. Results: 173 titles and abstracts were screened with 16 studies (158 judokas, 9 of which were female) included. All studies used 3D biomechanical analysis to assess Ukemi. Ukemi implementation produced reduced kinematic data in comparison to direct occipital contact, which was always below the injury threshold. Analysis of lower limb and trunk kinematics revealed variances in Ukemi between novice and experienced judoka. Whilst no significant differences were seen in neck flexion angles, hip, knee and trunk angle time plots revealed greater extension angles in experienced judokas. Conclusions: Ukemi is essential in preventing head and neck injuries; however, technique differs between experienced and novice judoka. Larger flexion angles of the hip, knee and trunk are seen in novice judoka, which correlate with increased kinematic data. The association of greater neck muscle strength with improved Ukemi is weak. However, a negative correlation was established between fatigue and breakfall skill by one study.

## 1. Introduction

Judo is an extremely popular martial art; it is an official Olympic and Paralympic sport, practised in over 200 countries worldwide [1,2]. It was first developed in 1882 by Professor Jugoro Kao, who created a more defensive style of martial art, going beyond sport, represented by three fundamental principles: (1) physical education; (2) contest proficiency and (3) mental training—all with the goal of being a better individual in society [2]. However, like with any sport, there are associated injury risks.

The aim of judo is to pin your opponent to the ground using grappling and throwing techniques [1,3]. Despite this more defensive style, an observational study from 2008 to 2016 recorded injury incidence rates in judo of 9.6 per 1000 min of exposure. Furthermore, in comparison to other combat sports such as boxing, taekwondo and wrestling, judo had the highest injury rate; this warrants investigation [4]. Common injuries include strains, sprains and contusions of the knee, shoulders and fingers [3,5]. Less frequent are injuries affecting the brain and spinal cord, such as acute subdural haematoma (ASDH); however, they are more prevalent in young and novice judokas [6,7]. In Japan, there has been a decline in judo participation in schools due to fear of severe injury. Over the past 30 years, approximately 300 students have been made disabled or comatose due to traumatic brain injuries (TBI) in Japan [8].

Judo consists of standing and ground fighting; however, the majority of injuries are mostly associated with standing, grappling and throwing moves [3,5]. Studies reporting the prevalence of injury suggest a greater association of injury to the uke (athlete being thrown) in comparison to the tori (throwing athlete) [9]. A 2016 study reported an injury rate of 43.8% in the uke population, as opposed to an injury rate of 25% seen in the tori population [10]. Osoto-gari (OS), Seoi-nage (SN) and Ouchi-gari (OU) are frequently used throwing techniques in high-level competitions. Amongst these techniques, SN is the most common throwing technique amongst judokas who sustained head and neck injury (42.9%) [11,12]. Head and neck injuries usually result from head impact onto the judo mat (tatami). It is believed that sudden rotational acceleration in the sagittal plane causes rupture of bridging veins resulting in TBI on impact [13,14,15,16].

Ukemi, or ‘safe falling’, is a breakfall motion technique that is emphasised in judo [17]. Before participating in combat practice (randori), judokas must perfect Ukemi [18,19]. Judokas learn variations of Ukemi to prepare them for randori; this includes sideways breakfall (Yoko Ukemi) [20], forward breakfall (Mae-Ukemi) [21], backwards breakfall (Ushiro Ukemi) and forward breakfall with a roll (Mae Maware Ukemi) [22]. However, this discipline has not prevented severe injuries from taking place [6,11,16], leading us to question the effectiveness of Ukemi. Hence, this study will assess the efficacy of Ukemi through an analysis of impact on the head and neck and evaluate whether improvements in technique can be made to reduce injury rates.

This systematic review aims to provide a clear understanding of the biomechanics of Ukemi, when implemented by experienced and novice judoka, and establish whether it is suitable protection from severe injury. A comprehensive review of the biomechanics of Ukemi has never been published. Several descriptive and observational studies of Ukemi have been undertaken in recent years. Many of these use advanced 3D kinematic analysis to understand breakfall motion associated with different throws in various populations [23,24,25,26,27,28,29,30,31,32,33,34,35,36,37,38,39]. Therefore, there is a need to review the current understanding of the biomechanics of Ukemi. From this, we will provide recommendations for further research topics and practical suggestions to prevent injury during practice.

## 2. Materials and Methods

### 2.1. Study Design

This systematic review was conducted and reported according to the Preferred Reporting Items for Systematic Reviews and Meta-Analyses (PRISMA) guidelines [40].

### 2.2. Inclusion and Exclusion Criteria

Articles were included if the following criteria were met: (1) reported biomechanical analysis of judo breakfall technique (Ukemi); (2) adult judoka population; (3) discussion of injury related to judo technique; (4) English papers.

Articles were excluded if the following criteria were met: (1) review and retrospective type articles; (2) computerised biomechanical models; (3) non-English papers; (4) judokas aged <18; (5) studies with no available abstract; (6) biomechanical analysis of the tori’s actions; (7) biomechanical analysis of other combat sports (jujitsu, karate and MMA). The inclusion-exclusion criteria are shown in Table 1.

### 2.3. Literature Search Strategy

An extensive literature search was performed using PubMed, Google Scholar, Science Direct and EMBASE from inception to April 2021. Articles on the biomechanics of judo techniques relating to injury in judo in the adult judoka population were selected and reviewed. A broad search was used as the literature on this topic is sparse; a more specific search produced too few results. The following Boolean combination of terms was used [41]: (‘Biomechanic*’OR ‘Biomechanical analysis’ OR ‘Biomechanical injury’ AND ‘Kinetics’ OR ‘Kinematics’ AND ‘Injury’ AND ‘Judo*’). MeSH subject headings were not used to narrow or broaden the search [42]. Due to the large difference between results in google scholar and the other databases, we chose to only use the first 200 results as sorted by relevance of Google Scholar ranking [43]. Two reviewers (RL and MP) independently performed the search to ensure the results of the literature search were identical. 

All publications were exported to Mendeley reference desktop Version1.19.4 (Elsevier, New York, NY, USA), where all duplicates were removed by the first author [44]. The papers were then imported to Rayyan, a web and mobile screening tool for systematic reviews (version 2016), where the reviewers collaboratively screened the articles [45]. A third reviewer was available if a consensus could not be reached. 

### 2.4. Literature Screening

Articles were screened in a step-by-step process in the order of title, abstract and full text, in line with the predetermined study criteria. The two authors (RL, MP) screened all titles and abstracts independently and selected potential studies. When all potential studies were agreed on by both authors, full texts were reviewed for articles that met the inclusion criteria or for papers that could not explicitly be excluded. Further articles were excluded if the full text revealed they did not meet the inclusion or met exclusion criteria. 

### 2.5. Data Extraction and Analysis

Data from eligible studies were extracted by one reviewer (RL) and independently verified by the second reviewer (MP). Data elements recorded included: author, year of publication, study design and basic participant characteristics (age, years; height, cm; and weight, kg), anthropomorphic test device (ATD) data, experience level (all judokas who had not reached their 1st Dan were considered novice) [46], throws being used and biomechanical assessment method. A narrative synthesis of the data was performed; a meta-analysis was not possible due to the heterogeneity between studies. 

### 2.6. Quality Assessment of Literature

A five-item study checklist developed by another systematic review analysing observational studies was used to assess the risk of bias in the individual studies [47,48]. The five items included were modified from the ‘strengthening the reporting of observational studies in epidemiology’ (STROBE) statement [49]. The items were (1) study setting, location and study period; (2) eligibility criteria, sources and methods of participant selection; (3) exposure definition and measurement; (4) study outcome definition and measurement; and (5) main results and precision (e.g., 95% confidence interval). Each study was analysed as having a low or high risk of bias for each statement. If reporting of said item was lacking or unclear, it would warrant a high risk of bias; low bias items were scored 1, high bias items were scored 0. Reviewers agreed that a total score > 3 was considered low bias, a score of 3 was considered moderate bias, a score of 2 was high bias and a score of 1 warranted exclusion due to extremely high bias. The explanation and elaboration article was used to give examples and methodology for examining articles [50]. The two researchers assessed quality independently and then resolved disagreements to form the current assessment. One study was excluded based on the risk of bias assessment.

## 3. Results

### 3.1. Study Selection

The online database search identified 1493 titles from PubMed, Google Scholar, Science Direct and EMBASE, respectively, the following number of publications were found in each database (7,19,37,1430). After duplicates were removed, 1403 remained; following the screening process, a total of 37 studies were assessed for eligibility, 16 of these [23,24,25,26,27,28,29,30,31,32,33,34,35,36,37,38] were included in the systematic review. Included studies were agreed upon by the two reviewers; a third reviewer was not needed to reach a consensus. Figure 1 represents the 2020 PRISMA flow diagram [51].

### 3.2. Study Characteristics and Data Extraction

A total of 158 judokas were included in this review, 9 were female and 149 were male, 54 were elite and 104 were novice judokas. All tori’s were elite judoka who knew how to correctly perform the throw. The age (years), weight (kg) and height (cm) mean and range were the following: 24 (18–65) years, 166.2 (164.2–184.7) kg and 68.6 (64.9–101.7) cm. Four studies (25%) investigated Ukemi comparing experienced and novice judokas; five studies (31%) investigated throws without Ukemi, using ATD dummies; four studies (25%) evaluated the Breakfall technique with no associated throw. Twelve (75%) studies evaluated breakfall with an associated throw (OS, OU, SN and Tai-otoshi (TO)); all studies (100%) used 3D biomechanical analysis, two studies (12.5%) evaluated EMG activity; two studies (12.5%) evaluated neck strength and four studies (25%) evaluated multi-planer motion. See Appendix A for outcome groupings and Table 2, Table 3, Table 4 and Table 5 for study characteristics and outcomes.

### 3.3. Study Quality Assessment

Of the 16 studies included, six [23,24,25,28,33,34] were high-quality studies and ten were moderate [26,27,29,30,31,32,35,36,37,38]. One study [39] was removed from the review due to the extremely high bias identified by the modified strobe criteria. The mean modified strobe score was 3.4, study quality ranged from 3 to 4. None of the studies described the study setting, location or period; 5 out of 16 (5/16) did not adequately describe the eligibility criteria and sources and methods of participant selection. All studies described exposure definition and measurement; 4/16 did not describe the study outcome definition and measurement sufficiently and 1/16 studies did not describe the main results with precision, see Figure 2.

## 4. Discussion and Suggestions—Narrative Synthesis

This review is the first to summarise the relationship between Ukemi and the impact of falls on the head and neck. Analysis of Ukemi biomechanics evaluates the effectiveness of Ukemi in reducing the impact of falls in judo as well as ascertaining the injury risk to the uke during a throw. Direct occipital contact onto the tatami (judo mat) produced greater acceleration and momentum values in comparison to when Ukemi was implemented. Kinematic data assessing impact was always below the injury threshold when Ukemi was performed. Further kinematic data revealed differences in breakfall technique between novice and experienced judoka. Whilst no significant differences were seen in neck flexion angles (NFA), hip, knee and trunk angle time plots revealed greater flexion angles in novice judokas.

### 4.1. Head Kinematics of Breakfall Motion

Direct impact of the head on the tatami is a major cause of head and neck injury [52]; it accounts for approximately 60% of ASDH in judo [38]. Impact responses of the head have frequently been described in terms of acceleration in cadaver and mechanical studies [53,54]. The current gold standard to assess head injury is the head injury criterion (HIC), determined by translational rotation. The US National Football League reported a HIC value of 250 for concussions [55].

#### 4.1.1. Translational (Linear) and Rotational (Angular) Acceleration

ATD studies revealed that occipital contact on the tatami during OS and OU induces high translational acceleration in the longitudinal direction [24,35,36]. However, Ukemi, following OS, dramatically reduced peak resultant translational acceleration (maximum value: 10.3 G), well below the HIC value for concussion [29,38]. This implies that Ukemi is a sufficient measure to prevent severe head injury.

However, the HIC does not consider rotational acceleration, which plays a role in head injury mechanisms [56,57]. Rotational acceleration is associated with traumatic brain injury, concussion, ASDH and axonal injury; therefore, it should be a variable in the head injury criterion [25,36,56,57,58]. ATD studies discovered that without Ukemi, the head experiences high rotational acceleration during OS (maximum value 5081.3 +/− 691.8 rad/s^2^) that would result in injury; translational acceleration of the same throw did not meet the HIC [24,36,38]. All throws assessed in this study (OS, OU, Seoi-Nage (SN) and Tai-Otoshi (TO)) produced peak resultant rotational acceleration values below the concussion limit (4500 (rad/s^2^) once Ukemi was applied [25,26,37]. A comparison of rotational acceleration in ATD and expert studies saw a five-fold reduction in acceleration [25,36,38]. In addition, the use of an under-mat significantly reduced translational acceleration in OS and OU [35,36] but had no effect on rotational acceleration [36]. This infers that an under-mat is a deficient shock absorber to reduce impact to the head [36]. Furthermore, the development of a new HIC that takes into account translational and rotational acceleration is needed.

Additionally, sagittal plane rotational acceleration has been linked to more severe outcomes than coronal and horizontal plane rotation [13,14,59], whilst ATD studies support this notion [24,36]; sagittal plane rotation may be reduced by Ukemi. Murayama found no difference between acceleration in the three planes during Ukemi implementation [38]. However, Ishikawa saw a greater acceleration in the sagittal plane following OS during Ukemi [26]. To assess whether Ukemi affects sagittal plane rotation, this relationship must be investigated further.

Our data on rotational acceleration highlights that the sudden backwards head rotation is a key component of severe head injury in addition to linear acceleration. Furthermore, correct Ukemi significantly reduces both acceleration values below limits relating to concussion; therefore, it is effective in protecting judoka from severe head and neck injury.

#### 4.1.2. Neck Muscle Strength

Neck muscle strength is thought to play a role in the prevention of head and neck injury [26,60]. It is assumed that greater neck strength equates to better control of neck muscles and, therefore, a better ability to prevent neck extension momentum and angular acceleration of the head [61]. However, following high-intensity judo practice, the angular acceleration of the head increased, but neck muscle strength did not decrease [26]. Therefore, fatigue may have a greater influence on Ukemi than neck muscle strength. This study was performed on experienced adult judokas; therefore, it is unlikely that neck muscle strength would change significantly due to fatigue. In the case of novice judokas, neck muscle strength may be a greater contributor to head injury. A 2016 review highlighted an association between reduced neck muscle strength and greater injury risk [60]. Therefore, future research should focus on assessing this relationship in judo. Regardless, the importance of rest incorporated in training should not be underestimated [26].

### 4.2. Neck Kinematics during Breakfall Motion

#### 4.2.1. Neck Flexion Angles (NFA)

From the studies included in this review, the most common measure of ‘impact’ to evaluate head and neck injury was the NFA time plots [27,28,29,30,31,32]. Neck flexion is taught as part of Ukemi to prevent head impact due to neck and head extension [61]. It is theorised that NFA will differ between elite and novice judoka as elite will have greater neck muscle strength and, therefore, a better ability to resist extension [26,27]. However, the majority of the literature suggests that towards the end of breakfall motion, there is no significant difference in NFA between novice and elite judoka [27,28,30]. One study found a minimal statistical difference between novice and experienced judoka. However, interpretation of these results warrants thought due to the small effect size [29]. No significant difference was seen in the head flexion angle between novice and elite judoka [28]. Whilst head and neck flexion may play a role in Ukemi, it may not be an adequate measure of Ukemi skill.

#### 4.2.2. Peak Angular Momentum of Neck Extension (PAMNE)

PAMNE is another measure to assess the likelihood of head and neck injury. It is implied that the greater the magnitude of PAMNE, the more likely injury will result, as there is a greater application of force to the head and neck [30]. A significant difference in PAMNE was seen between novice and experienced judoka when thrown by OS [30], indicating experienced judokas have a more advanced breakfall technique. Analysis of peak angular momentum (PAM) of the neck in the sagittal, frontal and horizontal planes demonstrated a greater PAM in the sagittal plane. However, horizontal and frontal plane momentum accounted for 30% of peak flexion momentum, demonstrating the multi-planar movement of the neck during OS. Hence, improvement of neck strength in all three planes may improve breakfall motion [32]. Currently, no significant association has been seen between neck muscle strength and impact [26,32]; further analysis is needed to determine this relationship.

### 4.3. Correct Ukemi

It is proposed that skill of breakfall technique can be determined by observation of upper and lower limb kinematics [62,63] in addition to head and neck kinematics. Avoidance of head contact on the tatami through Ukemi is key to preventing head and neck injury. However, the positioning of the upper and lower limbs may determine the likelihood of head contact by predicting disordered falling [23]. It is proposed that advanced judoka have greater control of their limbs during breakfall in comparison to novice judoka; hence, why more severe outcomes are associated with the novice population [6,7]. Three-dimensional analysis of hip, knee and trunk angle time plots identified variances in breakfall technique between experienced and novice judoka.

#### 4.3.1. Ushiro Ukemi

Ushiro Ukemi (backwards breakfall) is a motion where the uke strongly hits the tatami before the head reaches its lowest point [17,22]. This is deemed a protective mechanism for the cervical spine and head as it is believed to reduce the impact on the head and neck. Ushiro Ukemi, exhibited in OU, directly correlated with vertical velocity measures of the ukes head. At the beginning of the throw, vertical velocity increased until it reached its maximum value; the impact of the hand, forearm and trunk hitting the tatami induced a reduction in vertical velocity [23]. In contrast, in OS, the trunk and lower limbs hit the tatami after the head reached its lowest position—this is reflected in a greater vertical velocity [23] and rotational acceleration [38]. This signifies the importance of Ushiro Ukemi in preventing severe injury during backwards falls. Monitoring of Ukemi timing patterns would enable coaches to predict disordered falling and tailor training based on timing patterns.

#### 4.3.2. Hip and Knee Angle Time Plots

Observation of basic backwards breakfall with no associated throw showed little [29] to no difference [28] in hip angle time curves between novice and experienced judoka. However, breakfall of OS showed greater left hip flexion in novice judoka [30,31]. A straighter hip positioning was seen in more experienced judoka [30], suggesting that greater hip flexion during Ukemi following OS is associated with a greater risk of injury. The analysis of Ukemi with no associated throw may not be useful in identifying differences in novice and advanced judokas as the momentum of the uke may significantly differ with and without the uke being thrown.

In addition, observation of knee flexion angle time curves, basic backwards breakfall and OS demonstrate a significant difference between novice and experienced judoka. During Ukemi, experienced judoka show faster knee extension [28], as well as greater knee extension values throughout the entire motion [29,30]. Faster and greater knee extension may contribute to better control during the backwards fall. Hence, coaches may need to pay more attention to hip and knee kinematics during training.

#### 4.3.3. Trunk Angle Time Plots

It is proposed that trunk kinematics play an imperative role in breakfall technique as it demonstrates control during breakfall [64]. However, studies observing breakfall motion without an associated throw found no or minor differences in trunk flexion between novice and experienced judoka [27,28,29]. Both exhibited similar trunk flexion patterns, which remained stable towards the end of motion [27,28,29]. However, a slight increase in trunk flexion angle was observed in novice judokas over the same period [27,28]. The increase in flexion may indicate reduced trunk stability of the novice judoka; however, the effect size was small.

Further comparison of trunk flexion during OS demonstrated a greater flexed position in novice judoka; in comparison, experienced judoka maintained a straighter position throughout motion; however, trunk flexion was equivalent at the end phase of motion [30]. Evaluation of trunk kinematics during OS and OU, in elite judoka, revealed that the trunk and lower extremities hit the tatami after the head reached its lowest position in OS; the opposite was true for OU [23]. However, when these two throws were performed by novice judoka, no variances in trunk angle time plots were observed [31]. These findings infer that experienced judoka have greater trunk stability and, therefore, control over trunk flexion. The flexed position adopted by novice judokas is comparable to the ‘squat protective mechanism’, which enables the lower limb to absorb the potential energy and reduce the impact force. However, in OS, the lower extremities are unable to provide the breaking force using the squatting position as the body is mid-air whilst falling backwards [30]. Moreover, greater trunk flexion amplifies the risk of head and neck injury by increasing angular velocity [33]. Trunk extension acts as a protective mechanism; therefore, judokas should enhance core strength to be able to maintain trunk extension during Ukemi. In addition, it is suggested that strengthening the peripheral scapular and cervical muscles will support a stable relationship between the head and trunk, allowing judoka to have greater control during a fall [65]. However, the literature only implies a potential association between neck and trunk muscle function, which needs further investigation.

A comparative report analysing variances of body control of an older experienced and younger novice judoka found that after collision, the novice judoka was motionless, reflected by a torso angle of 0°. Whereas the experienced judoka rolled the trunk on collision, allowing for dispersal of energy, reflected by a torso angle of −25°. Whilst there were no differences in the speed of the centre of mass, the experienced judokas technique resulted in a reduction in ‘impact’. Further research is needed to assess the trunk roll technique in Ukemi and its association with reduced injury risk [34].

### 4.4. Clinical Implication

We can draw several clinical implications from this review; however, these should be interpreted with caution. All included studies were of high to moderate quality; however, heterogeneity of the methodology of the included studies made formulating concise conclusions challenging. The thresholds stated in this review are subject to change based on calculations of risk curves, which would change the interpretation of results [66]. Nevertheless, this review suggests that when Ukemi is performed correctly, a considerable reduction in impact on the head and neck is seen. This emphasises the importance of the practice in a sport that is changing. It is believed that adaption of the traditional Ukemi, known as ‘unorthodox Ukemi’, can elicit dangerous behaviours, such as head rolls, which call for the uke to purposefully land on their head as opposed to avoiding head contact [19]. One study in this review recognised an association between fatigue and greater impact on the head and neck. This study elicits that fatigue can reduce the control the uke has during the fall. Hence, the simple solution of coaches ensuring that judokas are not overly fatigued during practice can limit severe injury risk to the uke. An awareness of the coach’s responsibility towards the judoka as well as certification of coaches by the international judo federation is necessary to prevent injury [8]. This should especially be the case when coaching younger judoka, who are less experienced and require safeguarding. The association of a greater flexed position seen in novice judoka can be used by coaches to assess their skillset and predict dangerous falling patterns. Judoka who show these patterns, should not be encouraged to practice throws that are associated with severe injuries (OS). Furthermore, the practice of Ukemi should be introduced as early as possible in young judokas, as this is a preventative measure for injury. Further analysis of neck muscle strength is needed to examine the relationship with impact on the head and neck.

### 4.5. Limitations

There are some limitations of this review to consider. Firstly, biomechanical measurements of impact on the head and neck vary between studies (i.e., velocity, acceleration, momentum); whilst we grouped studies based on these measurements’ comparison of different variables of severe injury was not possible. Only nine participants were female; therefore, this data can be said to represent the male population but is not representative of female judoka, especially since differences between male and female judoka have been highlighted [67]. Lastly, the impact of a fall and, therefore, Ukemi response is likely to differ when the fall is associated with and without a throw; this review did not fully explore this topic.

### 4.6. Future Work

Future research should focus on Ukemi biomechanics during competitions and move away from comparative studies which involve breakfall motion analysis without an associated throw. The relationship between neck strength and the performance of Ukemi should also be explored. Only one study in this review directly measured neck strength as a variable. More research may suggest that greater neck strength could play a significant role in preventing head and neck injury. Similarly, only one study touched on the link between fatigue and its effect on the performance of Ukemi; this should be explored further as it is an easily applicable protective measure. Furthermore, the studies to date are not representative of the biomechanics of Ukemi in the female population; therefore, future research should include data from this cohort. In addition, setting up prospective studies analysing how Ukemi can prevent the number of injuries amongst judoka could further research.

Lastly, Ukemi practice can not only impact judo injury prevention but can also be implemented in the general population. Research can be directed at implementing Ukemi techniques to reduce the consequences of falls in the elderly population.

## 5. Conclusions

This review clarifies that Ukemi is essential in preventing severe injuries by reducing the ‘impact’ on the head and neck of the uke. The use of an under-mat did not prove to be adequate in reducing head impact. Good breakfall technique is associated with control of the whole body, including upper and lower limbs in addition to the head and neck. Small differences were seen in hip, knee and trunk angle time curves between novice and advanced judokas. Greater extension of the hip, knee and trunk in advanced judokas indicate that greater extension of the trunk and lower limb may act as a protective mechanism. Variance in lower body dynamic strength profiles between elite and novice judoka suggests greater strength provides better control. Timing patterns should therefore be analysed by coaches to predict disordered falling and highlight improvements that can be made. No study in this review found a correlation between greater neck strength and improved Ukemi. Improving neck strength is clearly not the simple solution to reducing ‘impact’ on the head and neck. However, more research is needed to assess this relationship. Fatigue has been shown to negatively impact Ukemi; therefore, the provision of appropriate breaks in training sessions and competitions may significantly reduce injury risk.

## Figures and Tables

**Figure 1 ijerph-19-04259-f001:**
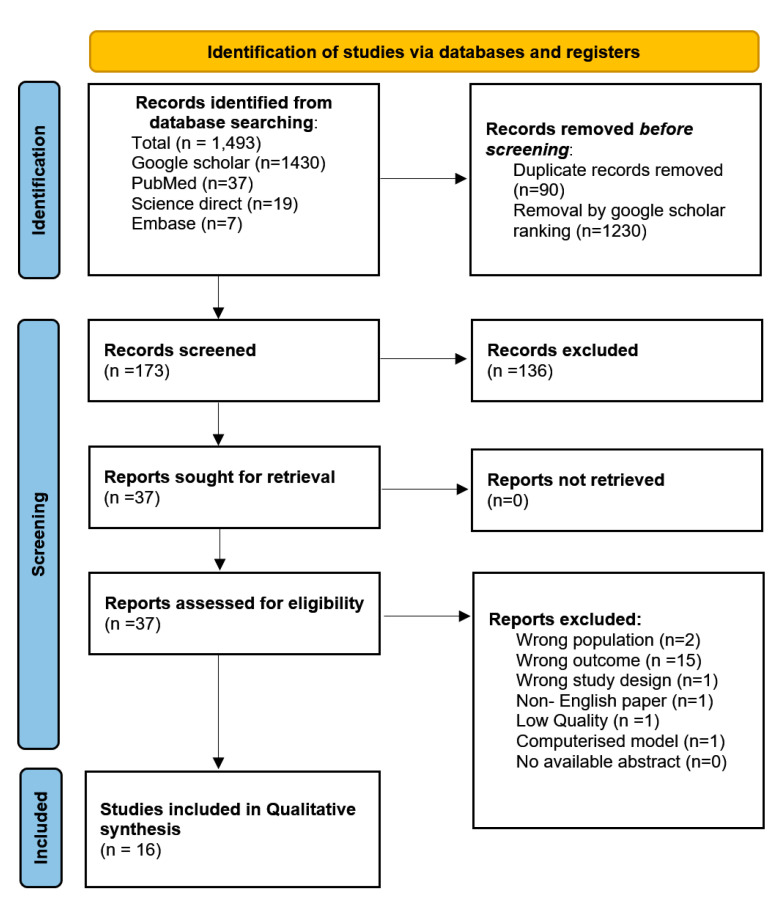
PRISMA flow diagram.

**Figure 2 ijerph-19-04259-f002:**
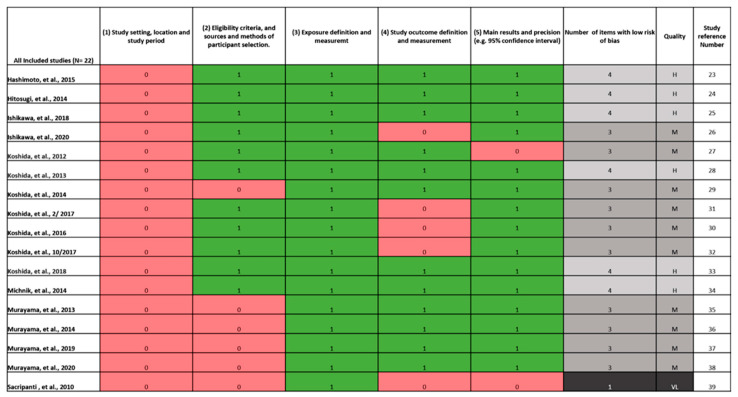
Modified STROBE criteria quality assessment. H = High quality. M = Moderate Quality. L = Low Quality. VL = Very Low Quality = Excluded.

**Table 1 ijerph-19-04259-t001:** Inclusion and exclusion criteria.

Inclusion	Exclusion
Reported Biomechanical analysis of judo breakfall technique (Ukemi) Adult judoka participants (>18 years old) Discussion of injury related to judo practice English papers	Review and retrospective articles Computerised biomechanical models Non-English papers Child participants (<18 years old) Studies with no available abstract Biomechanical analysis of the tori’s actions Biomechanical analysis of other combat sports

**Table 2 ijerph-19-04259-t002:** Study characteristics and outcomes.

Study Reference	Hashimoto, et al., 2015	Hitosugi, et al., 2014	Ishikawa, et al., 2018	Ishikawa, et al., 2020
**Study design**	Observational	Observational	Observational	Observational
**Participant characteristics: Number, gender, (Elite/Novice), (Tori/Uke), Age (years), Height (cm), Weight (kg)**	N = 8, Male 3, Elite Uke:(27, 184.7, 101.7) 5, Elite Tori: 3, (25, 169, 66) and 2, (27, 177, 93)	1, Male Elite Tori: (26, 177, 90) ADT dummy Uke, (NA, 175, 75)	9 Male 8, Novice Uke: (17.5, 173, 72.4) 1, Elite Tori: (20, 165.0, 70.0)	15 Male 14 Elite Uke: (19.4, 168.1, 77.5) 1Elite Tori: (18, 173.0, 74.0)
**Breakfall technique**	Exemplary Ukemi following OS and OU	No breakfall, of OS and OU	Basic Ukemi of OS, OU, SN, TO	Exemplary Ukemi of OS
**Biomechanical assessment method**	Vertical Velocity of the Uke’s head (kg m/s^2^)	3D Linear (G) and angular acceleration (rad/s^2^) of the uke’s head	3D Rotational acceleration of the ukes head (rad/s^2^)	3D angular acceleration of the ukes head (rad/s^2^) Neck muscle strength during forward & backward flexion (N)
**Measured Outcomes and key findings**	Vertical velocity OS > OU (204.82 +/− 19.95 > 118.46 +/− 63.62) *p* = 0.08 Vertical Velocity reduced when body surface area increased. In OS the head reached its lowest point before the trunk and lower limbs, the opposite is true for OU	Occipital head contact = large force in the longitudinal direction for linear acceleration and sagittal plane angular acceleration. Linear acceleration values in the longitudinal direction: OU > OS (41.0 +/− 2.6 G and 86.5 +/− 4.3 G) Angular acceleration values in the sagittal plane: OS > OU (3315 +/− 168 and 1328 +/− 201)	Max rotational acceleration generated: TO: 368.3, SN: 276.2, OS: 693.2, OU: 401.6 Rotational Acceleration: OS > OU > TO > SN	The maximum angular acceleration of the head immediately increased after high-intensity exercise (*p* < 0.01) Neck forward flexion strength increased (*p* < 0.05)
**Risk of bias**	Low	Low	Low	Moderate

**Table 3 ijerph-19-04259-t003:** Study characteristics and outcomes.

Study Reference	Koshida, et al., 2012	Koshida, et al., 2013	Koshida, et al., 2014	Koshida, et al., 2016
**Study design**	Observational	Observational	Observational	Observational
**Participant characteristics: Number, gender, (Elite/Novice), (Tori/Uke), Age (years), Height (cm), Weight (kg)**	10 Male 6, Elite Uke: (20.5, 171.9, 72.4) 4, Novice Uke: (20, 168.8, 68)	24 Male 11, Elite Uke:(19.9, 164.2, 70.1) 13, Novice Uke: (21.4,169.2, 68.6)	24 Male 11, Elite Uke: (19.9, 164.2, 70.1) 13, Novice Uke: (21.4,169.2, 68.6)	22 Male 12, Novice (21.3, 174, 71.3) 10 Elite, (19.9, 168, 70.1)
**Breakfall technique**	Basic Ukemi, no throw	Basic Ukemi, no throw	Basic Ukemi, no throw	Ukemi following OS
**Biomechanical assessment method**	Neck and Trunk flexion angle time curve (°) EMG amplitude (%) Of SCM, EO, RA	Head, neck-, trunk-, hip-, and knee-angle–time-curve profiles (°)	Peak Linear acceleration of the ukes head in the sagittal plane (g/s^2^) Neck, head, trunk, hip and knee flexion angle time profiles (°) EMG amplitude (%) Of SCM, EO, RA	Peak angular momentum of neck extension (kg m^2^s^−1)^ Neck, trunk, hip and knee flexion angles (°)
**Measured Outcomes and key findings**	Coefficient of multiple correlation (CMC) In neck and trunk values: (0.989 and 0.954), statistical significance (0.05) No significant difference between neck and Trunk flexion angle time curves and muscle activation between Novice and experienced judoka.	The results showed significant differences in knee (*p* < 0.001) and trunk (*p* < 0.005) flexion angle time curves, whereas no significant differences were found in head, neck, and hip kinematics between the novice and experienced judokas	No significant difference seen in mean peak linear acceleration in novice and elite judoka (1.69 +/− 0.48 g/s^2^ and 2.11 +/− 0.57 g/s^2^) *p* = 0.06 Neck, Hip and Trunk angles showed minimal differences between the groups A large significant difference was seen in knee extension movement. EMG activation patterns showed no significant difference between the two groups	Mean peak angular momentum of neck extension in the novice judokas (−1.29 ± 0.23) was significantly greater than that in the experienced judokas (−0.78 ± 0.28) No significant differences in the neck (*p* = 0.6) or right hip (*p* = 0.4) angles between the experienced and novice judokas pairwise comparison= significant differences in the trunk angle movement in OS (*p* < 0.001) significantly greater left hip flexion observed in the novice judokas in OS (*p* < 0.01) Greater knee flexion stability seen in experienced judokas (*p* > 0.005)
**Risk of bias**	Moderate	Low	Moderate	Moderate

**Table 4 ijerph-19-04259-t004:** Study characteristics and outcomes.

Study Reference	Koshida, et al., 2/2017	Koshida, et al., 10/2017	Koshida, et al., 2018	Michnik, et al., 2014
**Study design**	Observational	Observational	Observational	Observational
**Participant characteristics: Number, gender, (Elite/Novice), (Tori/Uke), Age (years), Height (cm), Weight (kg)**	13 Male 12 Novice, Uke: (21.3, 174, 71.3) 1 Elite, Tori: (38, 170, 73)	22 Male 21 Novice, Uke, (20.1, 170, 68.6) 1Elite, Tori, (41, 170, 65)	23 Male, 9 Female 31 Novice Uke, (20.9, 167, 64.9) 1 Elite Tori,	2 Male 1 novice Uke, (24, 183, 77) 1 Elite Uke, (65, 181, 84)
**Breakfall technique**	Ukemi following OS and OU	Ukemi following OS	Ukemi following OS	Basic Ukemi, no throw but knocked out of balance by 3rd party
**Biomechanical assessment method**	Mean peak angular momentum of neck extension (kg m^2^s^−1)^ neck, hip, Trunk, knee angle time plots (°)	Peak angular momentum of neck extension (kg m^2^s^−1)^ Neck flexion angles (°) Forward flexion neck muscle strength (N)	Peak neck angular momentum (kg m^2^s^−1)^ Trunk COM angular velocity (rad/s^2)^	Velocity of centre of mass Torso Angle of centre of mass (°)
**Measured Outcomes and key findings**	Mean peak angular momentum of neck extension in OS > OU: (1.29 +/− 0.23 And 0.84 +/− 0.29) *p* < 0.01 A significant difference was seen between OS and OU in neck, hip, and knee angle time plots (*p* < 0.01). No variances seen in trunk angles between OS and OU	Neck flexion angle increased until peak flexion, followed by abrupt extension at end. Neck flexion in OS is multidirectional, Peak angular momentum of the sagittal plane was greatest, but the Horizontal and frontal plane accounted for 30% of neck extension. No linear relationship between neck strength and angular momentum.	A significant correlation was seen between the trunk COM velocity and the peak neck angular momentum in novice judoka.	No difference was seen in the speed of the centre of mass between novice + elite. Differences were seen between Torso angles of novice and experienced judoka.
**Risk of bias**	Moderate	Moderate	Low	Low

**Table 5 ijerph-19-04259-t005:** Study characteristics and outcomes.

Study Reference	Murayama, et al., 2013	Murayama, et al., 2014	Murayama, et al., 2019	Murayama, et al., 2020
**Study design**	Observational	Observational	Observational	Observational
**Participant characteristics: Number, gender, (Elite/Novice), (Tori/Uke), Age (years), Height (cm), Weight (kg)**	1 Male 1 Elite, Tori, (26, 177, 90) ADT dummy Uke, (NA, 175, 75)	1 Male 1 Elite, Tori, (26, 177, 90) ADT dummy Uke, (NA, 175, 75)	1 Male 1 Elite, Tori, (33, 166, 82) ADT dummy Uke, (NA, 175, 75)	2 Male 1 Elite Tori, (29, 177, 90) 1 Elite Uke
**Breakfall technique**	No breakfall, of OS and OU With and without under-mat	No breakfall, of OS and OU With and without under-mat	No breakfall, of SN	Basic Ukemi, following OS
**Biomechanical assessment method**	Resultant Head acceleration (G) Head injury Criterion (HIC)	Peak translational (G)and rotational acceleration (rad/s^2^)	Peak linear (G) and angular (rad/s^2^) acceleration	Translational (G) and Rotational (rad/s^2^) acceleration
**Measured Outcomes and key findings**	Head acceleration in the longitudinal direction: OU > OS HIC values without under mat: OU and OS (1174.7 +/− 246.7) and (330.0 +/− 78.3) HIC values with under mat: OU and OS (539.3 +/− 43.5) and (156.1 +/− 30.4)	Translational acceleration: OU > OS, (130.0 +/− 13.2 and 74.4 +/− 9.8) Rotational acceleration: OS > OU (5081.3 +/− 691.8 and 1906.0 +/− 280.1) Translational acceleration was significantly reduced by use of an under-mat (*p* = 0.021) Rotational acceleration was not significantly reduced by use of an under-mat (*p* = 0.29)	Peak values of linear and angular acceleration did not significantly differ between 3 directional axes. High angular acceleration was observed (1890.1 +/− 1151.9) Increase in linear acceleration in the longitudinal direction and angular acceleration in the sagittal plane was not seen	No significant difference was seen in the three axis directions for both accelerations. Peak resultant rotational and translational accelerations (679.4 +/− 173.6 and 10.3 +/− 1.6) were significantly lower than previous ADT Study. (*p* = 0.0021)
**Risk of bias**	Moderate	Moderate	Moderate	Moderate

## Data Availability

The data presented in this study are available on request from the corresponding author.

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
