# Peer review of "A Systematic Review on the Biomechanics of Breakfall Technique (Ukemi) in Relation to Injury in Judo within the Adult Judoka Population"

_ijerph, 2022, doi:10.3390/ijerph19074259_

Round 1
Reviewer 1 Report
Dear Authors
You have done an interesting review of the specific field. It is nice to see that the authors performed an extensive search that is not years restricted.
However, some parts of the paper need to be addressed for greater clarity and some mistakes need to be corrected.
Abstract -
Line 24 - im-plementation / please check the whole text as you have - in a lot of words like that and it makes text hard to read. Amend accordingly
Line 32 / delete - in correlation /
Line 33 - I would suggest deleting the last part - by one study.
Introduction
Line 38 - Judo was founded in 1882 and not 1888 as you stated! Please delete reference 3 from this sentence as the 1888 year, the reference reports, is clearly a mistake. Please be critical of the literature you report and co-authors that are eminent judokas should notice that.
lines 44-45 / recent studies show lower injury rates than reported in this sentence and this needs to be added.
Lines 57-58 / This sentence needs to be rewritten as this is not what these studies reported. Be specific - one of these studies analyses ''drop seoi nage, which is totally different from ippon seoi nage. Amend
Line 63-65 / what about ushiro ukemi and mae maware ukemi? List all breakfalls and not just two. Amend
The last paragraph of the introduction - it needs to be clear that this is being reviewed on a sample of judokas. From what it is written it is not clear.
Methods:
Why just English papers? Elaborate
Why were case study papers excluded? Elaborate
Exclusion criteria 5-abstracts/ so you didn't check full papers?
You list 7 exclusion criteria while table 1 reports 8. correct
Table 1 reports the exclusion of under 16 and text under 18. What was it then?
Search strategy: why didn't you use keywords like breakfall, ukemi, etc? Elaborate
Figure 1 is of low quality and hard to read - correct
Rayyan - report what is this - software, platform, etc
Report, for how many references were consensus not reached and 3rd reviewer was needed.
Tables 2-5 / please add an explanation of all abbreviations and add units for variables like velocity, acceleration,
You wrote that case studies are excluded. So why do you report a study by Hitosugi et al. 2014 with 1 participant? This study should be excluded according to your criteria. Also studies by Michnik et al, 2014 and Murayama et al. 2013;2014; 2019 and 2020 are obvious case studies. Please elaborate and think about adapting your inclusion criteria.
Overall tables are too small and hard to read. Try to amend them to be clearer.
Line 353 - Thewy were not all good quality. Be specific according to your strobe findings. Amend accordingly.
What about guidelines for further research - this could complement and build on your limitation section. Also what about the impact of ukemi on youth and injuries could be mentioned.
In conclusion, you don't refer to any references. Please rewrite the sentence.
Line 391 - make conclusions just on what was proven - so writing ''Whilst neck strength may be important'' should be deleted as you just reported in the previous sentence that this was
References:
Please correct the use of all capital letters in the references.
Please check the following references as they are inconsistently referenced: 11, 20, 27, 36, 45, 60, 67, 70, 71
Reference 63 and 64 is cited twice! Correct
Overall an interesting review paper that still needs some more work. Therefore, I recommend major revision.
Kind regards
Author Response
Dear Reviewer,
Thank you very much for your time and valuable comments, which all have been considered and incorporated. The detailed list of responses is given below. We hope that the modifications and explanation will be acceptable for you.
Yours sincerely,
Rydzik, corresponding author
Abstract:
Point: ‘Line 24 - im- plementation / please check the whole text as you have - in a lot of words like that and it makes text hard to read. Amend accordingly’
Answer: Thank you for raising this point, I am unsure as to why this happened and I will change accordingly.
Point: ‘Line 32 / delete - in correlation /’
Answer: I have taken this into consideration and have deleted accordingly
Point: ‘Line 33 - I would suggest deleting the last part - by one study done’
Answer: I have taken this into consideration and have deleted accordingly
Introduction:
Point: ‘Line 38 - Judo was founded in 1882 and not 1888 as you stated! Please delete reference 3 from this sentence as the 1888 year, the reference reports, is clearly a mistake. Please be critical of the literature you report and co-authors that are eminent judokas should notice that.’
Answer: Thank you for raising this point. We acknowledge this and have taken care to correct this mistake. Reference 2 indicates that judo was founded in 1882 and also highlights the wholistic aspects of judo as a sport.
Point: ‘lines 44-45 / recent studies show lower injury rates than reported in this sentence and this needs to be added.’
Answer: Thank you for highlighting this, we acknowledge this and have taken care to include the most recent injury rates reported and update this reference. At the time I wrote the introduction I was unaware of the 2020 published study, which is a continuation of the article I used. In this most recent study competition injury rates were measured per 1000min of exposure. An injury rate of 9.6 per 1000 mins of exposure was seen. Furthermore, comparatively to other combat sports, judo was found to have the highest incident rate. We hope that this is acceptable to the reviewer.
Point: ‘Lines 57-58 / This sentence needs to be rewritten as this is not what these studies reported. Be specific - one of these studies analyses ''drop seoi nage, which is totally different from ippon seoi nage. Amend’
Answer: Thank you for raising this point. In reference 11, in the introduction it states that amongst judokas who sustain head and neck injuries, drop seoi-nage is the most common throwing technique (42.9%). In reference 12 it states that Osoto-gari, Ouchi-gari and Seoi-nage are some of the most commonly employed techniques in high level competitions. It was incorrect to state that these throws were the most commonly used techniques associated with head and neck injury.
Point: ‘Line 63-65 / what about ushiro ukemi and mae maware ukemi? List all breakfalls and not just two. Amend’
Answer: Thank you for raising this point, we have now included Ushiro Ukemi and Mae Maware Ukemi within the introduction
Point: ‘The last paragraph of the introduction - it needs to be clear that this is being reviewed on a sample of judokas. From what it is written it is not clear.’
Answer: Thank you for raising this point, we have re written this sentence to mention the involvement of experienced and novice judokas to make this point clearer. We hope this is satisfactory.
Methods:
Pont: ‘Why just English papers? Elaborate’
Answer: Thank you for raising this point, The reason why we only included English written paper or Papers that had been translated by a the author is because we did not believe that google translate would be a sufficient material to translate papers written in another language. Google translate commonly makes translation mistakes and we did not want to misinterpret results due to the language barrier. I hope this is satisfactory for the reviewer.
Point: ‘Why were case study papers excluded? Elaborate’
Answer: Initially we did not want to include case studies as we felt studies based off one subject would not have provided enough data to come to an accurate conclusion. However due to the limited research found on this topic and findings of the case study papers we decided to include case studies, as shown. However, we mistakenly did not remove this from the criteria. We have amended this by removing this from the initial exclusion criteria. We hope this is satisfactory to the reviewer.
Point: ‘Exclusion criteria 5-abstracts/ so you didn't check full papers?’
Answer: We included this in the exclusion criteria as part of our screening method was to read the abstract in order to assess whether this article was relevant. If it was deemed relevant or we were unsure whether it may be relevant these papers were included and sought for retrieval. The full paper was then assessed to determine relevance. However, there were no studies excluded due to an unavailable abstract. We hope this answer is satisfactory.
Point: ‘You list 7 exclusion criteria while table 1 reports 8. Correct’
Answer: We excluded animal models from the text as the biomechanics of Ukemi cannot be performed on animal models, therefore this criteria did not make sense. This was not excluded from table however has been corrected accordingly.
Point: Table 1 reports the exclusion of under 16 and text under 18. What was it then?
Answer: Thank you for raising this point, we were debating whether 16 or 18 should be the cut off point as judoka as young as 16 may have achieved their first Dan. However we eventually made the decision to exclude under 18 as they would legally be classed as children and this paper was focused on Ukemi in the adult population. I hope this answer is satisfactory to the reviewer.
Point: ‘Search strategy: why didn't you use keywords like breakfall, ukemi, etc? Elaborate’
Answer: We initially used a broad search strategy which highlighted these key words however when using this we came up with very little articles from the prospective databases. Therefore, we made a decision to exclude these in order to broaden our search and ensure that we did not miss any relevant papers. I hope this answer is satisfactory to the reviewer.
Point: ‘Figure 1 is of low quality and hard to read – correct’
Answer: We have improved the quality of figure 1.
Point: ‘Rayyan - report what is this - software, platform, etc’
Answer: Thank you for raising this point, we can see how it is unclear what type of platform Rayyan is. Therefore, we have amended this to state that it is an online web platform that allows papers to be easily grouped into various categories for systematic reviews. I hope this is satisfactory to the reviewer.
Point: ‘Report, for how many references were consensus not reached and 3rd reviewer was needed.’
Answer: A third reviewer was not needed for the two reviewers to reach a consensus, I have stated this in the results section.
Tables 2-5 / please add an explanation of all abbreviations and add units for variables like velocity, acceleration,
Answer: Thank you for Highlighting this. We have included the values for participant characteristic at the top of the table. We have stated what the abbreviations for Osoto-gari (OS), Ouchi-gari (OU), Seoi-nage (SN), are in the text before the table. Abbreviations which were not explained clearly are the following: COM, HIC, (G) is a unit measure for resultant acceleration measured by the head injury criterion equation. ATD anthropomorphic test device stated in text but will add to key to make clearer, EMG, SCM, EO, RA sternocleido-mastoid, external oblique and rectus abdominis muscle. We have included these in a table key
Point: ‘You wrote that case studies are excluded. So why do you report a study by Hitosugi et al. 2014 with 1 participant? This study should be excluded according to your criteria. Also studies by Michnik et al, 2014 and Murayama et al. 2013;2014; 2019 and 2020 are obvious case studies. Please elaborate and think about adapting your inclusion criteria.’
Answer: We have addressed this point above
Point: ‘Overall tables are too small and hard to read. Try to amend them to be clearer.’
Answer: Thank you for highlighting this, we have re-formatted tables 2-5 to make these clearer. We hope this is satisfactory for the reviewer.
Point: Line 353 - They were not all good quality. Be specific according to your strobe findings. Amend accordingly.
Answer: Thank you for highlighting this point, I believe this is in reference to line 358 ‘all studies were of good quality’. According to STROBE criteria 6 studies were of high quality however 10 studies, were of moderate quality. We used the word ‘good’ majority were of moderate to high quality. However we can understand how this statement may be misleading, therefore we have changed this statement accordingly.
Point: ‘What about guidelines for further research - this could complement and build on your limitation section. Also, what about the impact of ukemi on youth and injuries could be mentioned.’
Answer: Thank you for raising this point, in response we have built on my limitations section by suggesting what future research could be focused on. We have also highlighted what this papers finding will have on younger judoka as well as the clinical implications for the sport in general with regards to reducing the number of injuries.
Point: ‘In conclusion, you don't refer to any references. Please rewrite the sentence.’
Answer: We will remove this reference from the conclusion.
Point: ‘Line 391 - make conclusions just on what was proven - so writing ''Whilst neck strength may be important'' should be deleted as you just reported in the previous sentence that this was’
Answer: Thank you for highlighting this point. What we were intending to say by this line is that although no study included in the review found an association between greater neck strength and better breakfall technique, this was only one study. Therefore, to conclude that there is no association may be inaccurate as we believed more research on this topic was needed. However we agree that the conclusion should be based on our positive findings, therefore have chosen to discuss this in future research section. I hope this change is satisfactory to the reviewer.
References:
Point: ‘Please correct the use of all capital letters in the references.’
Answer: This has been amended accordingly
Point: ‘Please check the following references as they are inconsistently referenced: 11, 20, 27, 36, 45, 60, 67, 70, 7’
Answer: Thank you for pointing out these inconsistencies, in order to ensure these references were accurate we assessed each reference listed and ensured it was related to the statement in the text. Reference 11 was incorrectly sited in one place, regarding rotational acceleration, this was changed accordingly. Reference 20 is only sited once in regard to (Yoko ukemi) sideways breakfall, which is the correct reference. Reference 27 is sited in regard to findings on angle time plots. This paper observed breakfall without an associated throw, it found no difference in neck flexion angle time plots or in hip angle time plots, however found faster knee extension, this study, found a small difference between expert and novice judokas trunk angle time plots. The reports of this paper have been correctly sited. Reference 36 is related to Peak resultant rotational acceleration during Seoi-nage, this is correctly referenced in the text. Reference 45 is related the fact that the first dan is the minimum requirement to take part in judo competitions, therefore individuals who have not reached 1st dan are novice. I have changed this reference to the judo association web page as I couldn’t access previous paper to review. Reference 60: indicates that angular acceleration of the head have been linked to more severe outcomes, however this is sited in the text as related to the difference between Sagittal and horizontal plane rotation. Therefore, we will exclude this reference. Reference 67 describes Ushiro Ukemi, backwards breakfall, this has been correctly cited within the text. Reference 70 is related to the threshold stated in this review is subject to change based on risk calculation curves. Reference71 shows differences in schooling tendencies of men and women, I have not reference in the text however we have now used it to indicate why future research should focus on female studies and not just male. I hope these changes and explanations are satisfactory
Point: Reference 63 and 64 is cited twice! Correct
Answer: Thank you for highlighting this point, I will delete reference 64 accordingly.
Overall an interesting review paper that still needs some more work. Therefore, I recommend major revision.
Reviewer 2 Report
Dear authors,
First of all thanks for the opportunity to review your work. I believe it is very well made. Only some little remarks:
- a) many times along the text works are separated with an hyphen. Please correct.
- b) the criteria to only use the first 200 references of the Google Scholar. Does this option reduce the access to information? Because Google Scholar includes thesis, and you included only observational studies, is this an exclusion criteria?
Best regards
Author Response
Dear Reviewer,
Thank you very much for your time and valuable comments, which all have been considered and incorporated. The detailed list of responses is given below. We hope that the modifications and explanation will be acceptable for you.
Yours sincerely,
Rydzik, corresponding author
Point: ‘many times along the text works are separated with an hyphen. Please correct.’
Answer: Thank you for raising this point, We are unsure as to why this happened and I will change accordingly.
Point: ‘the criteria to only use the first 200 references of the Google Scholar. Does this option reduce the access to information? Because Google Scholar includes thesis, and you included only observational studies, is this an exclusion criteria?’
Answer: Thank you for raising this point, Our search strategy was extremely broad due to the fact that most databases except google scholar, provided very few relevant papers. Therefore, this gave us the understanding that there is limited research on this topic. Hence it is to go from finding approximately 30 relevant articles to over a 1000 is very unlikely therefore we deemed that the first viewing 200 articles would allow us to cast our net out far enough without excluding relevant articles. Google scholar sorts the papers based on relevance therefore the first 200 would be most relevant papers. Exclusion of thesis’s were not part of exclusion criteria. I hope this answer is satisfactory
Round 2
Reviewer 1 Report
Dear Authors,
Thank you for addressing my raised comments fully. The quality of the paper increased in my opinion. Overall I recommend acceptance and I congratulate authors on a job well done.
Kind regards
This manuscript is a resubmission of an earlier submission. The following is a list of the peer review reports and author responses from that submission.